

# Clinical and dosimetric correlation in terms of treatment response, bladder and rectal toxicities in cervical cancer patients treated with cobalt 60 high dose rate brachytherapy

Bharat Sai Makkapati[*], Srinivas Challapalli, Athiyamaan Mariappan Senthiappan, Johan Sunny Kilikunnel, Abhishek Krishna, Dilson Lobo, Vaishak Jawahar and Sourjya Banerjee[*]

Department of Radiation Oncology, Kasturba Medical College, Mangalore, Manipal Academy of Higher Education, Manipal, India
[*] These authors contributed equally to this work.

## ABSTRACT

**Background**. High dose rate (HDR) image-guided brachytherapy with Cobalt-60 isotope is a relatively recent approach. The aim of the study is to evaluate the clinical and dosimetric parameters in terms of tumour response, bladder, and rectal toxicity in patients undergoing Co-60 HDR brachytherapy.

**Materials and Method**. All patients were initially treated with chemoradiation (CT-RT) at our center or other referral centers with external beam radiation therapy (EBRT) for a dose of 45 Gy–60 Gy at 1.8-2Gy/fraction (including nodal boost) with concomitant chemotherapy with either cisplatin or carboplatin. Patients were then scheduled for brachytherapy within 1 week after completion of CT-RT and are assessed by local examination. Depending on local examination parameters at the time of brachytherapy they were eligible either for intracavitary brachytherapy (ICBT) or interstitial brachytherapy (ISBT).

**Results**. The complete response (CR) observed in stage I, II, III, IVA were 60%, 79.4%, 86% and 76.2% respectively. Complete response was seen in patients with mean EQD2 of 78.67 $Gy_{10}$, 83.33 $Gy_{10}$, 84.23 $Gy_{10}$, 85.63 $Gy_{10}$ in stages I, II, III, IVA respectively. 79.2% of cisplatin-treated patients and 87.5% of carboplatin-treated patients had a complete response indicating that patients treated with either chemotherapy had similar response rates.

**Conclusions**. According to results obtained from the study we conclude by saying that higher rates of complete response to treatment in cervical cancer is seen in patients with shorter overall treatment time (OTT), shorter interval between end of definitive CT-RT and beginning of brachytherapy and squamous cell histology. The study also noted the trend of increasing mean EQD2 to tumor with increasing stage for achieving complete response. Higher acute bladder and rectal toxicity is seen in patients who received EQD2 of >70-90$Gy_3$ and >70$Gy_3$ respectively. The study findings suggest that the clinical outcomes and the toxicities are clinically comparable with other radioisotope based HDR brachytherapy treatment.

Corresponding authors
Athiyamaan Mariappan Senthiappan, athiyamaan.ms@manipal.edu
Dilson Lobo, dilson.lobo@manipal.edu

## INTRODUCTION

Cervical cancer is the second most frequent malignancy among Indian women (*International Agency for Research on Cancer, 2020*). Patients with cervical cancer may experience vaginal discharge or bleeding, flank or abdominal pain, haematuria, or rectal bleeding (*Gupta, Aich & Deb, 2014*). They present at an advanced stage as a result of their poor socioeconomic position and inadequate utilization of screening facilities in developing countries. Standard treatment for stage IB to stage IVA cancers constitute external beam radiation therapy (EBRT) and concurrent chemotherapy, followed by brachytherapy (*Chargari et al., 2019*; *Gupta, Aich & Deb, 2014*). Brachytherapy (BT) is a form of radiotherapy in which a radioactive source is kept either directly into or adjacent to the tumor (*Chargari et al., 2019*). From manual pre-load low dose rate (LDR) to remote-controlled after-load high dose rate brachytherapy (HDR-BT), radioactive sources have undergone evolution (*Chargari et al., 2019*; *Gupta, Aich & Deb, 2014*). Brachytherapy delivered are either intracavitary brachytherapy (ICBT) or interstitial brachytherapy (ISBT) based on the anatomy and extent of residual disease after EBRT. BT has its own set of potential side effects such as bleeding, uterine perforation, bladder or rectal perforation, abdominal pain, diarrhoea, discomfort in the lower abdomen and lower back (*Gupta, Aich & Deb, 2014*). HDR image-guided brachytherapy with Co-60 isotope is a relatively recent approach. The aim of our study was to evaluate clinical and dosimetric parameters in terms of tumour response, bladder, and rectal toxicity in patients undergoing Cobalt 60 HDR brachytherapy.

## MATERIALS & METHODS

A longitudinal descriptive study was conducted at Kasturba Medical College Hospital, Attavar, Mangalore between September 2020 to June 2022 after obtaining the institutional ethics approval (Kasturba Medical College, Mangalore, Manipal Academy of Higher Education (deemed to be university) granted ethical approval to carry out the study within its facilities with Protocol No. IEC KMC MLR 12/2022/420).

Criteria of inclusion for the study were the histologically proven carcinoma cervix patients staged as per International Federation of Gynecology and Obstetrics (FIGO) stages IB–IVA and treated with concurrent chemoradiation therapy completely. A written informed consent was obtained from the patients to participate in the study and were provided a questionnaire to capture the demographic, basic medical information. According to the proforma, retrospective analysis parameters of the patients were extracted from medical records department and for prospective analysis, data was collected on case to case basis. Dosimetric parameters was extracted from SAGI plan treatment planning system (TPS).
All patients were initially treated with external bean radiation therapy (EBRT) at our center or other referral centers with a total dose of 45Gy–60Gy at 1.8-2Gy/fraction (including nodal boost) and concomitant chemotherapy with either cisplatin or carboplatin. Patients with paraaortic node positivity as per magnetic resonance imaging (MRI) assessment during the initial presentation received paraaortic extended field irradiation. Carboplatin was administered to patients who were unfit for receiving cisplatin due to creatinine clearance being <50 ml/min (Cockroft-Gault). The patients who did not receive concurrent chemotherapy along with EBRT were excluded from the study.

Patients were then scheduled for brachytherapy usually within 1 week after completion of CT-RT and are assessed by local examination (per vaginal, per speculum, per rectal) for any residual disease, vaginal status, cervical os, parametrial extent, rectal mucosal involvement. At the time of brachytherapy, patients were assigned to either ICBT or ISBT, based on the results of the local examination.

In ICBT, uterine tandem with flange and two vaginal ovoids are inserted under anaesthesia (spinal/general). In ISBT, interstitial needles are inserted using vaginal obturator in the Syed-Neblett template under epidural anaesthesia through transperineal route. First a guide needle is inserted into anterior cervical lip and remaining needles depending on extent of disease are inserted.

CT scan simulation was performed using applicators *in-situ* with 50ml of diluted contrast in bladder without the intravenous contrast. CT axial slices of three mm thickness were obtained.

The CT files were then transferred to SAGI TPS. Applicators and catheters were reconstructed in the brachytherapy planning module (Fig. 1). In accordance with Gyn GEC ESTRO contouring recommendations, we prescribed the high-risk clinical target volume (HR CTV) and contoured the organs at risk (OAR), such as the bladder and rectum. Inverse planning technique was followed with further optimization done manually by dragging isodose line using graphical tool. This was a trial and error process where in the dwell time/dwell position are adjusted to achieve optimal dose coverage to High risk clinical target volume (HRCTV) ($EQD_2 > 80Gy_{10}$) with constraints to bladder $D_{2cc}(EQD_2 < 85Gy_3)$ and rectum $D_{2cc}(EQD_2 < 75 Gy_3)$ (*Viswanathan et al., 2009*).

Clinical response were assessed at 3 months after brachytherapy completion. Using MRI, the response was assessed in compliance with RECIST criteria 1.1. After three months, acute toxicities of the bladder and rectal were evaluated and assessed according to RTOG acute toxicity criteria at end of 3 months (*Cox, Stetz & Pajak, 1995*).

The statistical analysis was carried out using the statistical tool SPSS version 22.0.

Continuous variables were presented as Mean ± Standard Deviation, Median, Min: Max, Q1:Q3. Frequency and percentage were used to represent categorical variables. Fisher's exact test was performed to see the association between two categorical variables. *P* value of less than 0.05 was considered significant.

## RESULTS

Study included 150 patients diagnosed with carcinoma cervix (Table 1), out of which six patients were excluded due to loss of follow-up and since they did not receive chemotherapy.

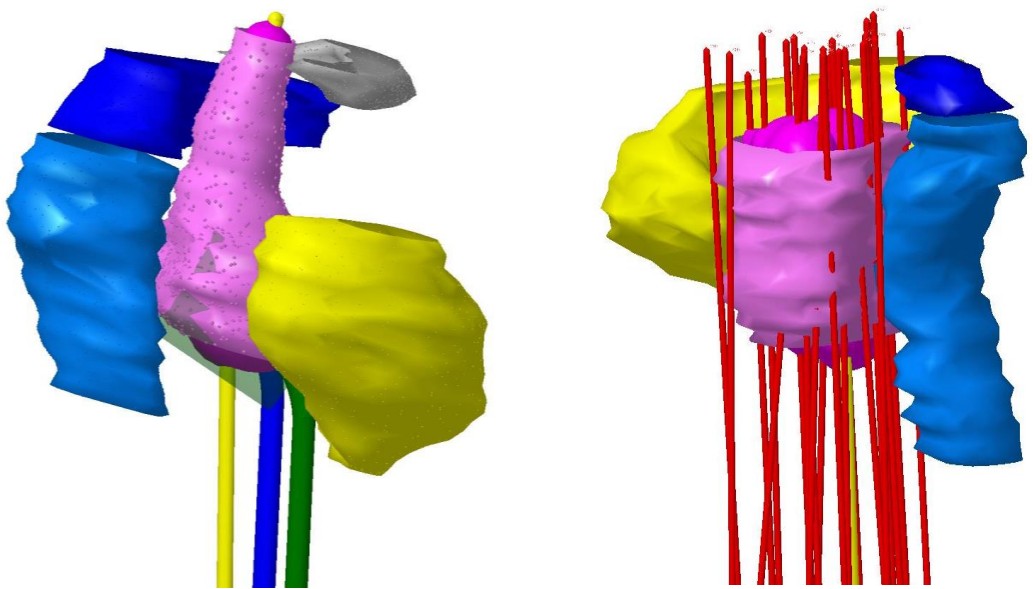

**Figure 1 Catheter reconstruction.**

The mean age was 53.63 ± 10.21 years. The complete response (CR) observed in stage I, II, III, IVA are 60%, 79.4%, 86% and 76.2% respectively. In the ulceroproliferative group, 113 patients (81.3%) had a complete response while 1 patient (33.3%) had a complete response in the endophytic group. CR was observed in 82.7% of patients in the squamous cell variant while 64.8% of patients with adenocarcinoma variant had a complete response. The $P$ value was found to be significant. The $P$ value was 0.0149, depicting that the histopathological variant of cervical cancer had a statistically significant effect on the outcome in our study, showing squamous cell carcinoma has the higher chance of clinical complete response when compared to adenocarcinoma.

Cisplatin was administered in 120 patients (83.3%). Carboplatin was administered to 24 patients (16.7%) who were deemed unfit for receiving cisplatin due to creatinine clearance being <50 ml/min (Cockroft-Gault). A total of 79.2 % of cisplatin-treated patients and 87.5 % of carboplatin-treated patients had a complete response (Fig. 2). It was found that there is no significant association between the type of chemotherapy received by the patient and clinical response at the end of three months. The $p$-value was 0.837, indicating that both carboplatin and cisplatin had similar outcomes and did not cause any significant alteration in the outcome.

Lymph nodal status in the initial MRI was compared with the radiological response (MRI) at the end of three months. Patients without significant regional lymphadenopathy had 79.6% complete response, patients with pelvic lymphadenopathy had a 80% complete response and patients with paraaortic lymphadenopathy had a 100% complete response at the end of 3 months in the current study. Patients with pelvic lymphadenopathy also showed a higher percentage of progressive disease (12.5% *versus* 10.2%) when compared to patients with no significant lympadenopathy.

**Table 1  Demographic details and observational data.**

| N | 144 |
| --- | --- |
| **Age (years)** | |
| Mean ± SD | 53.63 ±10.21 |
| Median | 53.00 |
| Range | 33.00: 82.00 |
| **Histopathology** | **n(%)** |
| Squamous cell carcinoma | 127(88.2%) |
| Adenocarcinoma | 17(11.8%) |
| **FIGO Staging 2018** | **n(%)** |
| I | 05(3.5%) |
| II | 68(47.2%) |
| III | 50(34.7%) |
| IV A | 21(14.6%) |
| **Lymph nodal status on initial MRI** | **n(%)** |
| Lymph node negative | 98(68.1%) |
| Positive pelvic node | 40(27.8%) |
| Positive paraaortic lymph node | 06(4.2%) |
| **Clinical morphology** | **n(%)** |
| Ulceroproliferative | 139(96.5%) |
| Ulcerative | 02(1.4%) |
| Endophytic | 03(2.1%) |

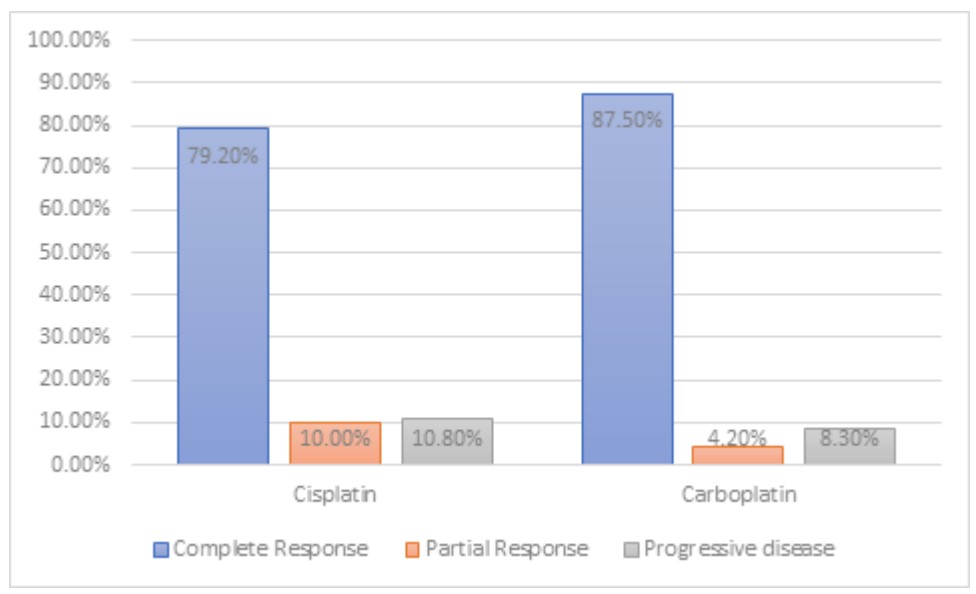

**Figure 2  Response assessment.**

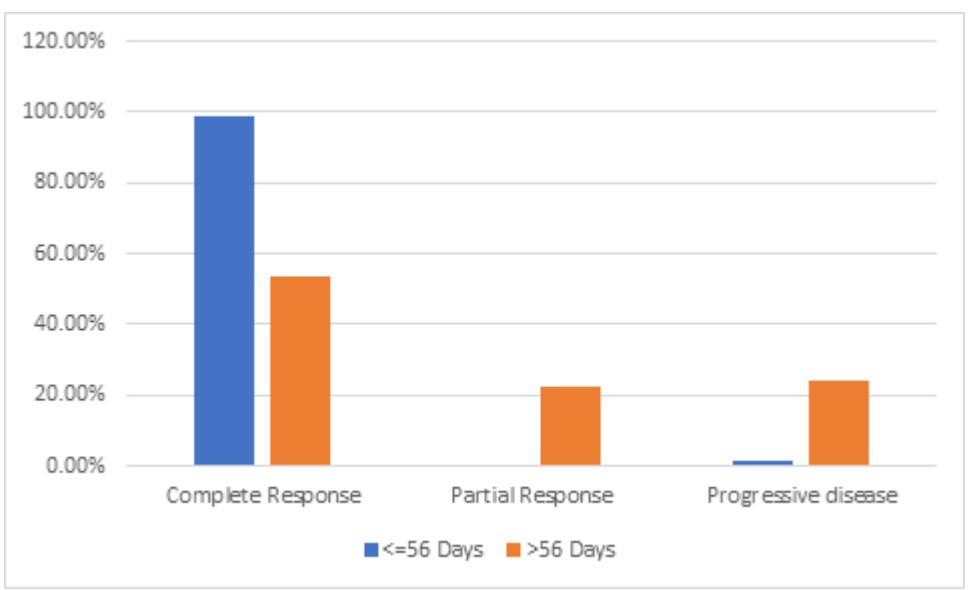

**Figure 3** Overall treatment time (OTT) and response.

Out of the 144 patients, overall treatment time (OTT) was within 56 days in 86 patients (59.7%) and 58 patients (40.3%) had a treatment duration of more than 56 days. Only 53.4% (31 patients) of the patients who completed the course of treatment in more than 56 days achieved CR, compared to 98.8% (85 patients) of the patients who completed the course of treatment in less than 56 days. (Fig. 3). Patients in more than 56 days arm had higher number of progressive diseases (24.1% *vs* 1.2%). A significant difference (*P* value < 0.001) was noted among the groups depicting that outcome worsens as overall treatment time increases. Patients whose brachytherapy was scheduled within 14 days after the end of definitive CT-RT had higher complete response rates (92.8 % *versus* 69.3%) and less percentage of progressive disease (7.2% *versus* 13.3%). The *P* value was < 0.001 which is statistically significant implying better outcomes when brachytherapy is scheduled within 2 weeks of completion of CT-RT.

The majority of patients (56.3%) did not show any bladder toxicity. A total of 31.2% had grade 1 toxicity, 11.1% had grade 2 and 1.4% had grade 4 toxicity of the bladder. 43% of patients showed ≤ grade 2 toxicities of bladder toxicity. Most patients received $EQD_2$ between 70-80 $Gy_3$. It is noted that majority of patients who received $EQD_2$ 70-90 $Gy_3$ had grade 1 toxicity (36%) when compared to other subsets (7%). The majority of patients received 70-75 $Gy_3$ as mean $EQD_2$ to rectum. The majority of the toxicities noted are of acute grade 1 and 2 (53.5%). A total of 43.8% of patients did not show any rectal toxicity. Grade 3 rectal toxicities were seen in three patients.

Complete response was seen in patients with mean $EQD_2$ of 78.67 $Gy_{10}$, 83.33 $Gy_{10}$, 84.23 $Gy_{10}$, 85.63 $Gy_{10}$ in stages I, II, III, IVA, respectively (Table 2). In stage I disease, a complete response was observed with a mean $EQD_2$ of 78.67 $Gy_{10}$, whereas progressive disease was observed with an $EQD_2$ of 77.0 $Gy_{10}$. In stage II mean $EQD_2$ for complete

**Table 2** Association of $EQD_2$ with response in different stage.

| Staging | Response | n | Mean ± SD ($Gy_{10}$) | Median | Min :Max |
|---|---|---|---|---|---|
| I | Complete response | 03 | 78.67 ± 2.89 | 77.00 | 77.00 : 82.00 |
| | Partial response | 00 | NA | NA | NA |
| | Progressive disease | 02 | 77.00 ± 0.00 | NA | NA |
| II | Complete response | 54 | 83.33 ± 3.08 | 83.00 | 76.00 : 90.00 |
| | Partial response | 07 | 84.29 ± 4.31 | 83.00 | 78.0 : 90.0 |
| | Progressive disease | 07 | 82.29 ± 2.36 | 83.00 | 77.00 : 84.00 |
| III | Complete response | 43 | 84.23 ± 4.52 | 83.00 | 77.00 : 96.00 |
| | Partial response | 03 | 84.67 ± 5.77 | 88.00 | 78.00 : 88.00 |
| | Progressive disease | 04 | 82.50 ± 0.58 | 82.50 | 82.00 : 83.00 |
| IV A | Complete response | 16 | 85.63 ± 3.05 | 85.50 | 80.00 : 90.00 |
| | Partial response | 03 | 88.67 ± 2.31 | 90.00 | 86.00 : 90.00 |
| | Progressive disease | 02 | 83.50 ± 0.71 | 83.50 | 83.00 : 84.00 |

response is 83.33 $Gy_{10}$ while mean $EQD_2$ of progressive disease was 82.29 $Gy_{10}$. In stage III disease complete response was noted with mean $EQD_2$ of 84.23 $Gy_{10}$ while progressive disease was seen with 82.50 $Gy_{10}$. In stage IVA patients with complete response had a mean $EQD_2$ of 85.63 $Gy_{10}$ whereas 83.50 $Gy_{10}$ was noted in progressive disease group. Five patients showed locoregional failure in absence of distant metastasis at the end of 3 months. Out of five patients with locoregional failure, four patients had local failure and one patient who had presented with node negative disease developed significant paraaortic lymphadenopathy. Ten patients showed evidence of distant metastasis mainly to liver.

## DISCUSSION

The aim of HDR brachytherapy is to administer high doses of radiation to tumor, to achieve greater control of disease with less toxicity to adjacent normal tissues. The advantage of brachytherapy comes from its dosimetric benefits, including the ability to deliver high and conformal doses to the site of disease with a rapid dose fall-off. The advantage of a Co-60 brachytherapy source is that it is a miniature source with longer half-life of 5.2 years when compared to the 73.8-day half-life of an Iridium 192 source.

The outcome of ICBT/ISBT depends on many factors—individual expertise, quality and timing of implantation, anatomy of treatment site, extent of vaginal packing, and type of source. This can lead to an excess of toxicity if the source is placed too close to a critical structure (*Chargari et al., 2019*).

A study by *Krebs et al. (2015)* which included patients with carcinoma cervix of FIGO stage IB1-IIIB, showed that 45% of patients had complete response and 55% patients had recurrent disease at end of 6 weeks post brachytherapy. In a study by *Pathy et al. (2015)*, which included patients with carcinoma cervix Stage IIB-IIIB who received definitive CT-RT followed by HDR brachytherapy, showed complete response of 76.0% at end of 3 months; this is similar to the outcomes from our study which showed 80.6% complete response at end of 3 months. Similarly, we observed 60.0%, 79.4%, 86.0% and 76.2% complete responses in stages I, II, III, and IVA, respectively. There has been a higher

complete response in stage III than in stages I and II. Patients with stage III with lymph nodal involvement received a nodal boost of additional 10 Grays to the involved nodes which helped to achieve a higher CR and with a higher EQD2 to stage III tumours. The other contributing factors for the favourable CR in stage III could be the tumour oxygenation which is independent of stage and histological type and grade. Another significant contributing factor could be the significant delay (>54 days) in referring patients for brachytherapy from nearby hospitals/other radiation centers after completion of EBRT in their center.

In a study by *Höckel et al. (1996)* including 100 patients demonstrated that tumor oxygenation in locally advanced cervical cancer is independent of its clinical stage, size, histological type, or grading. Radiation to the primary disease in patients with hypoxic tumors revealed significant disadvantages in the outcome (*Höckel et al., 1996*). In our study, among the patients with stage IB$_2$ cervical cancer, 40% (two patients) had progressive disease at end of 3 months, which can be attributed to tumor hypoxia. Tumor biology phenomenon must be taken into consideration when evaluating the effect of hypoxia on non-surgical anticancer treatments (*Höckel et al., 1996*). This may be a possible explanation for stage IB tumors (two patients) showing progressive disease in the present study.

While early outcomes are usually not important for survival variables and might be reflecting the tumors' sensitivity to RT, a study by *Katanyoo, Sanguanrungsirikul & Manusirivithaya (2012)* demonstrated that only 89 out of 367 patients (24%) who clinically had complete response were found to have disease recurrence with a median follow-up of 10.2 years (*Huang et al., 2010*. A study by *Pesee, Krusun & Padoongcharoen (2012)* showed one-year overall survival rates of 93.8% across all non-metastatic stages. In a study by *Rakhsha, Yousefi Kashi & Hoseini (2015)*, the 3 years DFS stages rates for patients who underwent EBRT (with or without concomitant chemotherapy followed by HDR Co 60 brachytherapy) were 85.7%, 70.4%, 41% and 16.6% for stages I, II, III, and IVA, respectively. With a median follow-up of 18 months, the 6-, 12-, 18-, and 24-month overall survivals were 98%, 86%, 75%, and 50%, respectively in a study by *Javadinia, Masoudian & Homaei Shandiz (2020)*. *Pesee, Krusun & Padoongcharoen (2012)* compared outcomes of HDR intracavitary brachytherapy using Ir 192 *versus* Co 60 source, which showed no statistically significant differences in 2 years OS (89.4% *vs* 91.6%). In a study by *Katanyoo, Sanguanrungsirikul & Manusirivithaya (2012)* including 423 patients with stages IIB–IVA, 367 (95.3%) patients had clinical CR. In our study, we observed 60%, 79.4%, 86% and 76.2% complete responses in stages I, II, III, and IVA respectively at end of 3 months after completion of brachytherapy. Thirty-two patients with pelvic lymphadenopathy (80%) had CR compared to six patients (100%) with paraaortic nodal involvement. Though the results look promising, this cannot be considered significant due to very low sample size (six patients) in paraortic nodal cases in the present study. The location, number and the size of the pelvic nodes at presentation may determine the nodal response to treatment.

While some reports showed no difference in survival outcomes between adenocarcinoma and squamous cell carcinoma (*Kilgore et al., 1988*; *Lee et al., 2006*), other authors reported poorer treatment outcomes for adenocarcinoma when compared to squamous cell carcinoma—especially in stages I and II (*Chen et al., 1999*; *Kleine et al., 1989*). A
retrospective study by *Kleine et al. (1989)*, (which included 268 squamous cell carcinoma patients and 144 adenocarcinoma patients of carcinoma cervix FIGO stages I-IV) there was decreased 5 year (53% *vs* 68%) and 10 year OS (42% *vs* 58%) rates in the adenocarcinoma group when compared to squamous cell carcinoma group. *Chen et al. (1999)* in a retrospective study consisting of 3678 patients of carcinoma cervix FIGO stage I-IV with either squamous cell carcinoma (91.4%) or adenocarcinoma (8.2%), showed decreased 5 year OS in adenocarcinoma group (66.5% *vs* 74.0%) when compared to squamous cell carcinoma group. Histopathological variants like squamous cell carcinoma and adenocarcinoma showed variable 1 year survival rates of 75.2% and 88.9% respectively (*Pesee, Krusun & Padoongcharoen, 2012*). In a retrospective analysis by *Katanyoo, Sanguanrungsirikul & Manusirivithaya (2012)*, 389 patients out of 423 patients had complete responses (CR), with squamous cell carcinoma and adenocarcinoma patients having 94.7% and 86.5 % CR, respectively. In our study, we noted 82.7% and 64.8% CR in squamous cell carcinoma and adenocarcinoma respectively at end of 3 months.

Overall survival and progression-free survival are significantly improved in the patients receiving chemotherapy concurrently with radiation in a study by *Peters et al. (2000)*. In a study by *Keys (1999)* 79% of patients in cisplatin plus radiation arm and 63% of patients in radiation alone arm had no evidence of disease after 36 months follow-up, concluded that, by adding weekly infusions of cisplatin to pelvic radiotherapy followed by hysterectomy significantly reduced the risk of disease recurrence and death in women with bulky stage IB cervical cancers. *Higgins et al. (2003)* concluded that carboplatin appears to be an effective alternative for cisplatin when combined with radiation therapy for cervix cancer upon evaluating in a study that included 31 patients with carcinoma cervix FIGO stage IB1-IIIB who underwent definitive radiation concurrently with carboplatin followed by brachytherapy and found evidence that 90% of patients had complete response rates. In our study, similar results were noted with the usage of carboplatin and cisplatin. 79.2% of cisplatin-treated patients and 87.5% of carboplatin-treated patients had a complete response in our study. All though 79.2% of cisplatin-treated patients and 87.5% of carboplatin-treated patients had a complete response , there is no significant association between the type of chemotherapy received by the patient and clinical response at the end of three months. The *p*-value is 0.837, indicating that both carboplatin and cisplatin had similar outcomes and did not cause any significant alteration in the outcome. This study does not prove that carboplatin can replace cisplatin in concurrent chemoradiation for cervix cancer. The durable response rate using concurrent carboplatin with radiation therapy in cervix cancer is unknown.

There were treatment delays due to lack of brachytherapy facilities geographically, lack of coordination between EBRT and brachytherapy scheduling, delays from referral institutions, hematologic toxicities (neutropenia, thrombocytopenia), anaesthesia fitness, and severe vaginal mucosal reactions. A correlation was found between OTT and pelvic control in a study of a series of several patients with stage I-III lesions with the cut off taken at 7 weeks (*Perez et al., 1995*). Prolongation of OTT resulted in a decreased pelvic tumor control rate of 0.85% per day for all patients, 0.37% per day in Stages IB and IIA, 0.68% per day in Stage IIB, and 0.54% for Stage III patients treated with >/ = 85Gy to point A

in a study by *Perez et al. (1995)*. *Tanderup et al. (2016)* also showed that Overall treatment time is an independent factor for local control. We observed a similar decrease in complete response rates if the treatment was not completed within 56 days (53.4% *vs* 98.8%) when compared to patients who completed treatment within 56 days.

A study by *Rakhsha, Yousefi Kashi & Hoseini (2015)* was conducted with 154 patients of carcinoma cervix who underwent Co 60 brachytherapy, comparing patients having a gap of 1 week and $\geq$ 3 weeks between definitive RT and brachytherapy. This study showed improved 3 year disease free survival (72% *vs* 49%) in patients who were scheduled for brachytherapy within 1 week of completion of external beam radiation. Similarly, in our study, patients whose brachytherapy was scheduled within 14 days of the end of definitive CT-RT had higher complete response rates (92.8% *versus* 69.3%) and lower progressive disease when compared to those whose brachytherapy was scheduled >14 days after definitive CT-RT (7.2% *versus* 13.3%). The *P* value was < 0.001, which is statistically significant—implying better outcomes when brachytherapy is scheduled within 2 weeks of completion of CT-RT.

Several studies showed relationship between local control and dose in carcinoma cervix. In a study from the Gustave Roussy Cancer Campus, crude control rates of >90–95% were seen when $CTV_{HR}D90 >85\text{-}87Gy_{10}$ (*Mazeron et al., 2015*). Mean total dose to point A EQD2Gy of $91.04Gy_{10}$ over stages IB2-IIIB showed 2 year and 5 year OS of 91.6% and 81.9% respectively (*Mazeron et al., 2015*). Dose escalation from 75Gy to 85Gy resulted in a 3% increase in local control in limited to intermediate size (20–30 cm$^3$), and 7% increase in large size (70 cm$^3$) to CTVHR (*Tanderup et al., 2016*). *Javadinia, Masoudian & Homaei Shandiz (2020)* showed a trend of reduced survival in patients with higher disease stages who were receiving lower doses to the tumor; however, the results were not significant. Similarly, our study showed that an increase in stage of the disease (from stage I to IVA) was associated with a trend of increasing mean $EQD_2$ $Gy_{10}$ for obtaining complete responses. Compared to the complete responses, the mean $EQD_2$ $Gy_{10}$ for the progressive disease was lower in each stage.

The incidence of metastases to other organs was 56%, with most frequent sites being lung, abdominal cavity, liver, and gastrointestinal tract, and the incidence of clinically apparent lymph node involvement (*Fagundes et al., 1992*). In our study, out of the five patients with locoregional failure, four patients had local failure and one patient had significant paraaortic lymphadenopathy. In our study, 10 patients showed evidence of distant metastasis to liver (four patients), lung (two patients) and non-regional lymph nodal metastasis (supraclavicular and inguinal lymph nodes).

In a study by *Ntekim et al. (2010)* that included 70 patients of carcinoma cervix patients who underwent definitive chemo-radiation followed by brachytherapy and were then followed up for 3 months, 56% of patients were identified to have $\leq$ grade 2 acute toxicities of bladder. Patients frequently complained of cystitis (40%), increased frequency of urination (40%) and urinary urgency (60%) (*Ntekim et al., 2010*). We observed approximately 43% of $\leq$ grade 2 toxicities of bladder in our study. No grade 3 acute genitourinary toxicities were seen and two patients had grade 4 acute toxicities in our study.

According to a study by *Kumar et al. (2017)* patients with acute gastrointestinal toxicities ≤ grade 2 had proctitis (56%) and diarrhoea (58%). Grade 2 acute toxicities of the rectum were seen in 38 percent of patients in a study by *Ntekim et al. (2010)* where 70 patients of carcinoma cervix who underwent definitive chemo-radiation followed by brachytherapy were followed up for a 3 month period. In *Kumar et al. (2017)*, 3% of patients had grade 3 diarrhoea among those patients who underwent Co 60 HDR intracavitary brachytherapy. We observed ≤ grade 2 toxicities of rectum in 54% of patients in our study. 2.1% (three patients) had grade 4 acute rectal toxicities in our study. Out of three patients with grade 4 acute rectal toxicities, one patient was on anticoagulant and antiplatelet therapy due to an underlying heart disease, which is a possible confounding factor in the present study.

## CONCLUSIONS

According to results obtained from the study we conclude that higher rates of complete response to treatment in cervical cancer is seen in patients with shorter OTT, shorter interval between end of definitive CT-RT and beginning of brachytherapy and squamous cell histology. In our study we also noted trend of increasing mean $EQD_2$ to tumor with increasing stage for achieving complete response. Higher acute bladder and rectal toxicity is seen in patients who received $EQD_2$ of > 70-90$Gy_3$ and >70$Gy_3$ respectively. The study findings suggest that the clinical outcomes and the toxicities are clinically comparable with other radioisotope based HDR Brachytherapy treatment. Long follow-up periods are necessary for the evaluation of response, long-term or late toxicity, and survival outcomes, which could be a limitation of our study. We also did not have an uniform pattern of scheduling between EBRT and brachytherapy in patients referred from several other centers in the city for brachytherapy at our center.

## ACKNOWLEDGEMENTS

Authors would like to express sincere thanks to Ms. Supriya Acharya for statistical evaluation, Ms. Mamatha, Ms. Pooja and all the department of Radiation Oncology KMC Mangalore for their active involvement and participation this study.

### Funding

The authors received no funding for this work.

### Competing Interests

The authors declare there are no competing interests.

### Author Contributions

• Bharat Sai Makkapati conceived and designed the experiments, performed the experiments, analyzed the data, authored or reviewed drafts of the article, and approved the final draft.

- Srinivas Challapalli conceived and designed the experiments, performed the experiments, authored or reviewed drafts of the article, and approved the final draft.
- Athiyamaan Mariappan Senthiappan conceived and designed the experiments, performed the experiments, analyzed the data, authored or reviewed drafts of the article, and approved the final draft.
- Johan Sunny Kilikunnel analyzed the data, authored or reviewed drafts of the article, and approved the final draft.
- Abhishek Krishna analyzed the data, authored or reviewed drafts of the article, and approved the final draft.
- Dilson Lobo conceived and designed the experiments, performed the experiments, analyzed the data, prepared figures and/or tables, authored or reviewed drafts of the article, and approved the final draft.
- Vaishak Jawahar analyzed the data, prepared figures and/or tables, and approved the final draft.
- Sourjya Banerjee conceived and designed the experiments, performed the experiments, analyzed the data, authored or reviewed drafts of the article, and approved the final draft.

### Human Ethics

The following information was supplied relating to ethical approvals (i.e., approving body and any reference numbers):

Kasturba Medical College, Mangalore, Manipal Academy of Higher Education (deemed to be university) granted ethical approval to carry out the study within its facilities (Protocol No. IEC KMC MLR 12/2022/420.

### Data Availability

The data is available in the Supplementary Files.

### Supplemental Information

Supplemental information for this article can be found online at http://dx.doi.org/10.7717/peerj.17759#supplemental-information.

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
