# Peer review of "Clinical and dosimetric correlation in terms of treatment response, bladder and rectal toxicities in cervical cancer patients treated with cobalt 60 high dose rate brachytherapy"

_PeerJ, doi:10.7717/peerj.17759_

## Round 0.1 · original submission · Minor Revisions

· Academic Editor

Minor Revisions

Reviewers provided valuable insights and offered impactful feedback on this manuscript. Kindly incorporate their suggestions and carefully address all comments in your manuscript. This impact is particularly meaningful in the context of cervical cancer research.

·

Basic reporting

Some sentences in the manuscript could have been rephrased better. Also, a flow in the content could be improved, in that one issue is discussed before going to the next (eg. after all discussion on response rates, toxicity to be discussed).

Experimental design

Being an observational study, it adds to the available literature on brachytherapy with Co-60 based HDR equipment. The necessary information has been provided.

Validity of the findings

Short term outcome reports including toxicity and response rates have been discussed. While not very novel, the findings add to the available literature on outcomes of carcinoma cervix treated with Co-60 based brachytherapy.

Additional comments

I request the authors to please look into the following shortcomings:
1. The article requires some editing to enhance the flow, and the language could be improved.
2. Clearly state the eligibility criteria for recruiting the patients for the stud; clarify whether patients who did not receive concurrent chemotherapy were excluded.
3. Line 126: the numbers stated appear to be incorrect; kindly correct the same.
4. Since the study describes experience with Co-60 based RT, it would be better to compare your findings with literature reporting Ir-192 or other isotope based treatment. A concluding remark on these lines (stating that your findings suggest that the outcomes are clinically comparable with other radioisotope based HDR treatment equipment) will justify the title.

Reviewer 2 ·

Basic reporting

It is a well conducted study with excellent results. However some things needs to be clarified.
1. There are many spelling mistakes which needs to be rectified (for eg: Its Syed and not Syedd)
2. Table 1 depicting demographic data needs additional information like chemotherapy details, OTT since these parameters have been discussed in the article.

Experimental design

Experimental design is well written.

Validity of the findings

Some findings need explanation such as
1. How do you explain higher complete response in Stage 3 compared to stage 1 or 2? What could be the contributing factors?
2. Treatment response depends on external beam radiation dose also. How will you justify the role of HDR brachytherapy alone in that case?
3. Whether patients with paraaortic node at presentation received extended field radiation? its not been mentioned in the article.
4. One patient with paraaortic recurrence had initial paraaortic node or is it a new finding?
5. What could be the cause for complete response in paraaortic nodal positive cases compared to pelvic nodal positive cases?
6. Why there is wide variation in EBRT doses?
7. What could be the reason for higher response in patients who received Inj Carboplatin compared to Inj Cisplatin?

Additional comments

Nil

---

## Round 0.2 · accepted · Accept

· Academic Editor

Accept

The authors have conducted great work on this manuscript, incorporating feedback from reviewers to enhance the clarity and depth of the content. They have successfully elucidated the role cobalt 60 dose impact in brachytherapy. This manuscript now offers a clearer and more insightful understanding of the mechanisms at play, thanks to the thoughtful integration of the reviewers' suggestions.

·

Basic reporting

Reasonable.

Experimental design

Being an observational study, there are no serious lacunae.

Validity of the findings

Acceptable, though only short term outcomes have been reported.